# Effect of Varying Dietary Crude Protein Level on Milk Production, Nutrient Digestibility, and Serum Metabolites by Lactating Donkeys

**DOI:** 10.3390/ani12162066

**Published:** 2022-08-13

**Authors:** Yuanxi Yue, Li Li, Manman Tong, Shuyi Li, Yanli Zhao, Xiaoyu Guo, Yongmei Guo, Binlin Shi, Sumei Yan

**Affiliations:** Inner Mongolia Key Laboratory of Animal Nutrition and Feed Science, College of Animal Science, Inner Mongolia Agricultural University, Hohhot 010018, China

**Keywords:** crude protein, milk production, nutrient digestibility, serum metabolites, lactating donkeys

## Abstract

**Simple Summary:**

Donkey milk, a functional food, can be used as a milk replacement for newborn nutrition, due to its similar chemical composition to maternal breast milk and hypoallergenic property, and may be useful in the prevention of hypercholesterolemia and atherosclerosis. However, donkey milk yield is very low and cannot satisfy the demands of the market. Some research on dairy cows showed that increasing dietary crude protein levels can increase milk yield and milk component yields. Therefore, this study explored whether increasing dietary crude protein levels could promote the milk production of lactating donkeys. The results showed that increasing crude protein levels could improve milk performance and nutrient digestibility of lactating donkeys. The key finding of this study was that a diet containing 14.2% crude protein supplied adequate protein to improve milk production in lactating donkeys.

**Abstract:**

Donkey milk is considered as a functional food due to its high levels of whey protein, and can be used in newborn nutrition, due to the nutritional similarities with human milk and its hypoallergenic property. However, donkey milk yield is very low and little is known about improving donkey milk yield by nutrition manipulation. The effect of dietary crude protein (CP) levels on milk production, nutrient digestibility, and serum metabolites was investigated in the current study. Twenty-four lactating donkeys were randomly assigned to one of the following three CP content diets: 15.3% (HP), 14.2% (MP), and 13.1% (LP) of dry matter, respectively. The experiment lasted for 10 weeks, with the first two weeks being used for adaptation. The results showed that milk yield and yields of protein, lactose, solid-not-fat, total solid, and contents of protein, total solid and milk urea nitrogen in the HP and MP groups were higher than the LP group. No significant changes were observed in dry-matter intake, contents of milk fat, lactose or solid-not-fat. The feed conversion ratio, milk protein synthesis efficiency, and the digestibility of dry matter, crude protein, ether extract, acid detergent fiber, neutral detergent fiber, calcium and phosphorous in the HP and MP groups were greater than the LP group. Serum total protein, albumin and urea nitrogen concentrations decreased, while concentrations of non-esterified fatty acids and β-hydroxybutyrate increased in the LP group compared with the HP and MP groups. In conclusion, the diet containing 14.2% CP supplied an adequate amount of protein for improving milk production in lactating donkeys, but milk production was not further increased by feeding the donkeys more than 14.2% CP.

## 1. Introduction

Donkeys are used for meat and milk production, as well as to obtain skin gelatine used in Asian traditional medicines. Nowadays, donkey milk has attracted more and more attention, due to the nutritional and functional characteristics [1]. Donkey milk can be used in newborn nutrition, because it has a rich chemical composition that is very similar to human breast milk [2,3,4]. Some studies showed that donkey milk could be used successfully as a succedaneum to hypoallergenic milk powder for people suffering from food allergies, especially in infants suffering from a cow’s milk protein allergy [5,6]. Clinical studies have clearly indicated that donkey milk may be very useful in the prevention of hypercholesterolemia and atherosclerosis from an immunological point of view [7,8]. In comparison with ruminant milk, donkey milk generally contains higher levels of whey protein [9] and lysozymes, which are known for their antioxidative and antibacterial properties [10,11].

Donkey milk yield remains at a low level, as it is limited to the special physiological structure without milk cistern of the mammary glands [12]. Salari et al. [13] reported that the average milk yield of donkeys was about 1–1.5 kg collected per day per head, which was lower compared to dairy cows (more than 25 kg/day) [14,15,16] and buffalo (4.8–7.2 kg/day) [17]. Milk performance, including milk yield and milk composition, is influenced by the diet, climatic conditions, lactation and animal health [18,19,20,21]. It is highly necessary for the donkey industry to improve donkey milk yield by nutrition manipulation. However, the majority of previous researchers have focused on dairy cows, and few studies on donkeys are available. Some research on dairy cows showed that increasing dietary crude protein (CP) levels can enhance milk and protein yields [14,15,16]. Sevi et al. [22] and Sinclair et al. [23] also reported on the effect of dietary protein concentration on improving milk yield in ewes and sows, respectively. Katongole et al. [24] and Colmenero et al. [16] suggested that the CP digestibility showed linear or quadratic increases with increasing dietary CP content for dairy cows. The effect of nutritional manipulation on milk performance of lactational donkeys is unclear. Thus, the objective of this study was to investigate the effects of dietary protein levels of dry matter (DM)) on milk production, nutrient digestibility, and serum metabolites by lactating donkeys in order to provide a basis for improving yield and quality of donkey milk. 

## 2. Materials and Methods

The experiment was conducted in the experimental farm of Inner Mongolia Agri-cultural University (Hohhot, China). All procedures were approved by the Technical Committee for Laboratory Animal Sciences of the Standardisation Administration of China (SAC/TC281), and performed under the national standard Guidelines for Ethical Review of Animal Welfare (GB/T 35892-2018).

### 2.1. Animals, Diets and Experimental Design 

In a completely randomized experimental design, a total of 24 lactating Dezhou donkeys (34 ± 4.9 days of milk, 2.52 ± 0.72 kg/d of milk yield, 2.8 ± 0.6 parity, 239 ± 24 kg of live weight) were randomly assigned to the following three treatments with eight replicates each: high crude protein (HP; 15.3% CP), medium crude protein (MP; 14.2% CP) and low crude protein (LP; 13.1% CP) on a DM basis. Diets were formulated to be isoenergetic and the content of digestible energy was 12.40 MJ/kg of DM for all diets, with a concentrate to forage ratio of 30:70. The ingredients and composition of the experimental diets are presented in Table 1. The experiment period lasted for 10 weeks, and included 2 weeks for adaptation (pretrial period) and 8 weeks for data and sample collection (experimental period).

### 2.2. Feeding Management, Sample Collection and Measurements

The donkeys were individually housed in a single stall with their foal and fed twice daily at 07:00 h and 14:00 h. Water was supplied ad libitum. For each donkey, feed intake was estimated daily based on the amount of feed offered and refused. The amounts of diets offered was adjusted daily on the basis of the previous day’s intake to ensure 5% refusal in the morning. The lactating donkeys were routinely machine milked by a wheeled trolley milking machine twice daily, at 10:00 h and 17:00 h. Foals were physically separated from the dams 3 h before milking and maintained visibility with the female donkeys. Feed intake (DMI) and milking yield (MY, kg/day) were recorded daily. Milk yield was calculated as follows: estimated milk yield (EMY, kg/day) = milking yield (kg/day) × 4 [25]; solids-corrected milk (SCM, kg/day) = {(12.3 × milk fat (%) content of non-standard milk +6.56 × solid-not-fat (%) content of non-standard milk−0.0752)} × Estimated milk yield (kg/day) [26], energy-corrected milk (ECM, kg/day) = 0.327 × EMY (kg/day) + 12.95 × fat (kg/day) + 7.65 × protein (kg/day) [27]. Feed conversion rate was expressed as follows: EMY (kg/day)/DMI (kg/day), SCM (kg/d)/DMI (kg/day) and ECM (kg/day)/DMI (kg/day). Milk protein synthesis efficiency = {EMY (kg/day) × milk protein (%)}/{DMI (kg/day) × CP level (%)} [28].

Milk samples were collected on the last day of every week. Morning and afternoon milk samples were blended and analyzed for protein, fat, lactose, total solid (TS), solid-not-fat (SNF) and urea-nitrogen (MUN) contents, using an automatic milk analyzer with mid-infrared waveband (MilkoScan Combifoss 7; Foss Electric, Hillerød, Denmark). Samples of total mixed rations were collected twice weekly and stored at −20 °C for chemical analysis. Feed samples were dried in a forced-air oven at 65 °C for 48 h, and then ground through a 1 mm screen, after being triturated for subsequent analyses of DM (method 930.15), CP (*N* × 6.25; method 984.13), ether extract (EE, method 920.39), calcium (Ca) and phosphorous (P, method 935.13), according to AOAC (2000) [29]. Neutral detergent fibre (NDF) and acid detergent fibre (ADF) were determined according to the methods described by Van Soest et al. [30] with an Ankom 220 Fiber Analyser (Ankom Technology Corp., Fairport, NY, USA) and were expressed as inclusive of residual ash. The content of hydrochloric acid insoluble ash (AIA) in the diets was determined using a gravitation technique, modified from the technique described by McCarthy et al. [31].

Fecal samples were collected from the rectum of each donkey every 8 h a day on 6 consecutive days in week 8 of the experimental period. Fecal samples from each donkey were thoroughly mixed and were divided into 2 portions. One pooled portion was added with 10% sulfuric acid for subsequent nitrogen analysis. Other fecal samples were stored frozen until the end of the experimental period and dried in a forced-air oven at 65 °C for 48 h, and then ground through a 1-mm mesh screen to determine the DM, EE, NDF, ADF, and AIA content. Apparent total-tract nutrient digestibility was calculated using AIA as an internal marker, based on the concentration of AIA in the diet and feces [32].

Blood samples were collected into test tubes (Corning Incorporated Costar, Corning, NY, USA) from the jugular vein of all experimental donkeys before morning feeding on the 7th day of week 8, and then centrifuged at 2500× *g* for 15 min to separate the serum. Serum was stored at −20 °C and determined for biochemical parameters. Concentrations of total protein (TP), albumin (ALB), blood urea nitrogen (BUN), glucose (GLU), total cholesterol (TC), alkaline phosphatase (ALP), calcium (Ca), phosphorus (P), β-hydroxybutyrate (BHBA), and non-esterified fatty acids (NEFA) in serum were determined by using an automatic biochemical analyzer (7020 Automatic Analyzer, 713-0002, HITACHI, Tokyo, Japan).

### 2.3. Statistical Analysis

All statistical analyses were performed in SAS software (version 9.4, SAS Institute Inc., Cary, NC, USA). Treatment effects of DMI, yields of milk, feed conversion rate, milk protein synthesis efficiency, milk composition, and yield of milk protein, fat, lactose, solid-not-fat and total solids were analyzed using the PROC MIXED procedure as follows:Yijkm = μ + Ci + Wj + Ci Wj + bXjk + Sim + εijkm,
where Yijkm = the dependent variable; μ = overall mean; Ci = fixed effect of dietary CP levels, Wj = fixed effect of lactation week (weeks 1, 2, 3, 4, 5, 6, 7 and 8), Ci Wj = effect of the interaction between diet treatment and lactation week, bXjk = effect of covariate (week 0, the observations during the 2 weeks of pretrial period served as covariates for the corresponding experimental period), Sim = a random effect (individual donkey); εijkm = residual error. Means were separated by using the PDIFF option in the LSMEANS statement. The nutrient digestibility and biochemical indicators in blood at week 8 were analyzed using the GLM procedure. Multiple comparisons were carried out in the Duncan’s test. Differences with *p* ≤ 0.05 were considered significant and differences with 0.05 < *p* < 0.10 were demonstrated a tendency toward statistical significance, and adjusted *p*-values were used in the PROC MIXED procedure.

## 3. Results

### 3.1. Milk Yield and Components

Donkeys fed the HP and MP diets had a greater MY, EMY, ECM, SCM, and feed conversion rate (EMY/DMI, ECM/DMI and SCM/DMI) and milk protein synthesis efficiency than donkeys fed the LP diet (*p* < 0.05), with no significant differences among the HP diet or MP diet. Milk protein, TS and MUN contents in donkeys fed HP and MP diets were higher (*p* < 0.0001, *p* = 0.013, *p* = 0.039) than the LP diet, with no significant differences among the HP diet or MP diet. Milk SNF content in donkeys fed HP tended to increase (*p* = 0.079) compared with the LP diet. Milk fat and lactose contents were not affected by dietary CP level. Donkeys fed HP and MP diets had significantly higher yields of milk protein, fat, lactose, TS and SNF than donkeys fed the LP diet (*p* < 0.0001), with no difference between HP diet and MP diet.

### 3.2. Feed Intake and Nutrient Digestibility

As shown in Table 2, there was no significant effect on DMI among the three groups during the total experiment period (*p* = 0.862). As shown in Table 3, the total tract apparent digestibility of DM, CP, EE, ADF, NDF, Ca and P in donkeys fed the LP diet was lower than those of HP and MP diets (*p* < 0.05), with no difference between the HP diet or MP diet.

### 3.3. Serum Metabolites

As shown in Table 4, compared to the LP group, the concentrations of TP, ALB and BUN were significantly increased (*p* = 0.010, *p* = 0.011, *p* = 0.002) in the HP and MP groups, whereas the concentrations of NEFA and BHBA were decreased (*p* = 0.002, *p* = 0.013), with no significant differences among the HP group or MP group. The GLU concentration in the HP group was significantly increased (*p* = 0.018) compared to the LP group, but the value for the MP group did not differ from those for either HP or LP groups. There were no significant differences in serum ALP, TC, Ca and P concentrations among the three groups.

## 4. Discussion

To the best of our knowledge, there are few studies that investigate the effects of dietary protein level in lactating donkeys on milk production. In a study [33] on female donkeys during the later gestation period, increasing dietary CP levels can increase the contents of dry matter, protein and most single amino acids in the colostrum of donkeys. The current study found that increasing dietary CP proportion from 13.1% to 14.2% meant that the milk yield significantly increased, and suggested that increasing CP levels of diets had a positive effect on milk yield in lactating donkeys. However, the milk yield was increased in value but the improvement was not statistically significant when the dietary CP level increased from 14.2% to 15.3% in the present study. Meanwhile, most of the increase in milk protein, fat, lactose, SNF and TS production, protein and TS content occurred in 13.1% to 14.2% of dietary CP, but only a slight improvement was observed in 14.2% to 15.3% of dietary CP. 

As was the case in the current study, Broderick et al. [34] reported in the experiment setting 15.1%, 16.7% and 18.4% dietary CP levels for dairy cows and no further improvement in yield of milk and protein beyond 16.7% CP. Katongole et al. [24] also found milk yield increased in a linear manner with dietary CP levels increasing from 14.1% to 17.7%, but increasing the dietary CP level beyond 17.7% did not result in improved milk yield, and noted a similar pattern of increase in milk protein yield and content with milk yield. Barros et al. [35] also reported a linear increase in yield of milk components when cows were fed diets with CP levels from 11.8% to 16.2%, and the substantial increases appeared with 11.8% to 14.4% dietary CP, and the increases were minimal with dietary CP beyond 14.4%. Furthermore, Leonardi et al. [36] provided the observation that milk protein content decreased when dietary CP content was increased from 16.1% to 18.9%. These publications demonstrated there was a threshold for dietary CP levels that affected milk composition and yield, and the threshold was 14.2% in the current study. In the current study, milk fat and lactose contents were unaffected by dietary CP level, which was in accordance with studies by Katongole et al. [24], Tebbe et al. [37] and Barros et al. [34]. Lactose has a regulatory role in the control of the osmotic pressure of milk, and lactose content generally remains constant [38]. Based on the results above, the production of milk protein, SNF and TS was increased due to the improved yield of milk and contents of milk protein, SNF and TS, and the production in milk fat and lactose was increased, attributed to the improved milk yield. The milk yield and milk component yield increased with CP increase but only up to a certain point, suggesting that a suitable CP requirement for milk production exists, which may be related to dietary nutrient digestibility, feed conversion ratio and protein utilization efficiency. Our outcomes showed that the feed conversion ratio and milk protein synthesis efficiency in the HP and MP groups were enhanced compared to the LP groups. Barros et al. [34] indicated that increasing dietary CP levels gave rise to increased feed efficiency (Milk/DMI), and the result was similar with our study. We also noted that milk protein synthesis efficiency decreased numerically when dietary CP content was increased from 14.2% to 15.3% in the current study. The fact that the LP group had lower milk protein synthesis efficiency was due to the lower milk yield and protein yield, while the decreased milk protein synthesis efficiency of the HP group was caused by the higher dietary CP intake and slight increased milk protein yield compared with the MP group.

Aguerre et al. [39] and Lee et al. [40] did not find an effect of diets ranging from 15.3% to 16.6% and 14.8% to 16.7% CP on DMI in dairy cows, respectively. Oliveira et al. [41] also reported that DMI was unaffected by dietary CP levels in horses. In the present study, we observed no difference in DMI between the dietary protein levels, which might be caused by the isoenergetic diets.

In some trials using dairy cows, increasing dietary CP levels was conducive to improving the apparent total tract digestibility of DM, OM, CP, ADF, and NDF [40,42,43]. In a study on horses, high CP diets enhanced the digestibility of DM, CP, ADF and NDF [41]. Additionally, the responses of CP digestibility in the present study were similar to those reported by Almeida et al. [44] and Karlsson et al. [45], who observed an increase in CP digestibility with increasing dietary CP levels in horses. The observations were consistent with the results of the present study, suggesting that HP and MP diets can improve nutrient digestibility to increase absorption of nutrients in donkeys. The outcomes further supported increased dietary CP as an important driver for increased milk yields, which may be attributed to the improvement of nutrient digestibility. Furthermore, Oliveira et al. [41] considered that the positive impact of dietary protein levels on NDF and ADF digestibility was affected by microbial growth and fermentative activity in horses. According to Julliand et al. [46], supplementing soybean meal can stimulate the growth of the proteolytic microbial population in the cecum and colon of horses. These responses suggested a positive effect of the high CP level on improving the cellulolytic bacteria in the cecum and colon. Therefore, we speculated that the enhancement of NDF and ADF digestibility caused by the increasing CP level was probably related to the improvement of bacteria in the hindgut. However, the result for bacteria in the hindgut of donkeys was not determined in this study, and it deserves further investigation.

BUN, a waste product of protein metabolism in the liver, can be used to evaluate nitrogen excretion and utilization efficiency in animals [47]. Broderick et al. [48] reported that a high concentration of blood urea was indicative of inefficient use of dietary CP by dairy cows. For lactating animals, blood urea rapidly equilibrates with milk [49], giving rise to a high correlation between blood urea and milk urea [48]. Colmenero et al. [16] reported a linear increase in BUN and MUN concentrations in dairy cows with increasing CP level from 13.5% to 19.4%. In a study on mares and goats, high CP levels in diets resulted in increased BUN concentrations [50,51]. Some other studies also confirmed greater MUN concentrations from cows fed high CP diets compared with those fed low CP diets [24,43]. In the current study, we observed higher concentrations of BUN and MUN in the HP and MP groups compared to the LP group. Although no significant differences in the concentrations of BUN and MUN between the HP group and MP group were found, we still found an increase in value for the HP group compared to the MP group. The outcomes indicated that dietary CP content exceeding 14.2% probably resulted in a reduction in the protein utilization efficiency. Additionally, Xia et al. [52] reported that low CP content reduced BUN concentration, but did not improve nitrogen efficiency in cows. Katongole et al. [24] also showed that MUN concentration was decreased, but milk protein synthesis efficiency was not increased when dietary CP content was decreased from 15.1% to 14.1% in dairy cows. Therefore, overfeeding donkeys with CP also reduces profit margins because of the relatively high cost of protein supplements and the poor efficiency of nitrogen utilization by donkeys fed HP diets, and the LP diet may result in the lack of protein intake and reduction in milk production.

Serum parameters play an important role in reflecting the healthy and nutritional status of animals [34]. A negative energy balance is reflected by increased BHBA and NEFA concentrations [53]. Increased NEFA and BHBA concentrations indicate that the body lacks energy and will promote steatolysis to energize itself. Adewuyi et al. [54] indicated that increased concentrations of BHBA and NEFA, along with decreased concentrations of GLU in blood plasma, also showed that the metabolic status of animals was impaired. In the present study, LP diet increased serum NEFA and BHBA concentrations compared to the HP diet, which showed that more body fat was mobilized due to the decreased nutrient digestibility, meaning that energy intake failed to meet the requirements for milk production in LP diet. This also suggested decreased protein utilization efficiency in the LP group. TP and ALB were the indicators of nutrition status in the body, and increased TP and ALB concentrations indicated that the animals received adequate nutrition [55]. In the present study, serum TP and ALB concentrations were increased with increasing dietary CP content. The similar results were observed in dairy cows [56] and goats [51]. Based on the results above, we found that 15.3% and 14.2% dietary CP levels can satisfy the nutritional requirement of milk production, and the 15.3% dietary CP level even resulted in oversupply of protein. Furthermore, some studies reported that increasing dietary CP content can promote mammary gland development of dairy heifers [57] and rats [58], respectively. Velázquez-Villegas et al. [59] indicated that high CP content in diets improved the prolactin concentration of rats, which stimulated mammary gland development and supported lactation, providing amino acids to the gland through the sodium coupled neutral amino acid transporter 2 (SAT2) for the synthesis of milk proteins. Future research is required on the effects of dietary CP level on mammary gland development in donkeys.

Additionally, in the current study, we used the equations from dairy cows to calculate the parameters (SCM, ECM, feed conversion rate and milk protein synthesis efficiency) of donkeys, in spite of the existing limitations of applying equations across species. A related study in donkeys has not been reported, and needs to be explored further.

## 5. Conclusions

In conclusion, increasing dietary CP level from 13.1% to 14.2% can improve milk production, feed conversion rate and milk protein synthesis efficiency of lactating donkeys, but milk production was not further increased by feeding the animals more than 14.2% CP.

## Figures and Tables

**Table 1 animals-12-02066-t001:** Composition and nutrient levels of basal diet.

Items	HP	MP	LP
Ingredients (% of DM)			
Millet straw	51.86	51.95	51.96
Alfalfa	15.87	15.88	15.84
Corn	15.65	17.67	19.79
Soybean meal	8.24	7.91	6.97
Corn gluten meal	3.27	1.63	0.33
Flax cake	1.92	1.92	1.92
Corn germ meal	0.64	0.48	0.49
Bran	0.48	0.47	0.63
NaCl	0.44	0.44	0.44
CaCO3	0.38	0.38	0.38
CaHPO4	0.76	0.76	0.76
Premix ^1^	0.50	0.50	0.50
Total	100.00	100.00	100.00
Chemical composition (% of DM)			
DE (MJ/kg) ^2^	12.39	12.40	12.41
CP	15.30	14.20	13.10
EE	2.96	2.94	2.95
NDF	50.78	50.75	50.87
ADF	30.14	30.10	30.06
Ca	1.18	1.20	1.19
P	0.37	0.36	0.36

HP, high crude protein; MP, medium crude protein; LP, low crude protein; DE, digestible energy; DM, dry matter; CP, crude protein; EE, ether extract; NDF, neutral detergent fiber; ADF, acid detergent fiber; Ca, Calcium; P, Phosphorus. ^1^ Provided per kg of premix: vitamin A 1,200,000 IU, vitamin D 250,000 IU, vitamin E 3000 IU, Fe 4.0 g, Cu 1.6 g, Zn 12 g, Mn 12 g, I 72 mg, Se 60 mg, Co 100 mg. ^2^ DE was a calculated value according to the Chinese Feed Ingredients and Nutritional Value Table (30th edition).

**Table 2 animals-12-02066-t002:** Effects of dietary crude protein levels on DMI, feed conversion rate, milk protein synthesis efficiency, milk yield, content and yield of milk components of lactating donkeys.

Items	HP	MP	LP	SEM	*p*-Value
DMI (kg/day)	7.38	7.39	7.49	0.163	0.862
Milking yield (kg/day)	0.85 ^a^	0.82 ^a^	0.70 ^b^	0.016	<0.0001
EMY (kg/day)	3.41 ^a^	3.27 ^a^	2.80 ^b^	0.064	<0.0001
ECM (kg/day)	1.73 ^a^	1.65 ^a^	1.39 ^b^	0.041	<0.0001
SCM (kg/day)	1.84 ^a^	1.73 ^a^	1.50 ^b^	0.043	<0.0001
EMY/DMI	0.46 ^a^	0.44 ^a^	0.37 ^b^	0.010	<0.0001
SCM/DMI	025 ^a^	0.24 ^a^	0.20 ^b^	0.006	<0.0001
ECM/DMI	0.23 ^a^	0.22 ^a^	0.19 ^b^	0.006	<0.0001
Milk protein synthesis efficiency	0.059 ^a^	0.061 ^a^	0.054 ^b^	0.0014	0.003
Milk components					
Fat (%)	0.28	0.27	0.25	0.012	0.146
Protein (%)	1.89 ^a^	1.87 ^a^	1.82 ^b^	0.010	<0.0001
Lactose (%)	7.11	7.08	7.07	0.016	0.256
SNF (%)	8.94	8.93	8.89	0.017	0.079
TS (%)	9.20 ^a^	9.18 ^a^	9.13 ^b^	0.018	0.013
MUN (mg/dL)	41.06 ^a^	39.89 ^a^	33.98 ^b^	1.253	0.039
Milk component yield					
Fat (g/day)	9.58 ^a^	8.83 ^ab^	6.98 ^b^	0.580	<0.0001
Protein (g/day)	64.44 ^a^	60.84 ^a^	50.93 ^b^	1.500	<0.0001
Lactose (g/day)	243.34 ^a^	232.27 ^a^	197.29 ^b^	5.415	<0.0001
SNF (g/day)	304.53 ^a^	291.50 ^a^	247.78 ^b^	6.746	<0.0001
TS (g/day)	313.54 ^a^	299.68 ^a^	254.91 ^b^	7.201	<0.0001

HP, high crude protein; MP, medium crude protein; LP, low crude protein; SEM = standard error of least square means; ^a, b^ means within the same row not followed by the same letters are significantly different at *p* < 0.05.

**Table 3 animals-12-02066-t003:** Effects of dietary crude protein levels on apparent total-tract nutrient digestibility of lactating donkeys.

Items	HP	WP	LP	SEM	*p*-Value
DM (%)	73.92 ^a^	73.63 ^a^	69.88 ^b^	0.969	0.041
CP (%)	82.21 ^a^	80.28 ^a^	74.40 ^b^	1.299	0.034
EE (%)	58.32 ^a^	57.47 ^a^	50.67 ^b^	1.419	0.003
NDF (%)	45.17 ^a^	43.42 ^a^	34.03 ^b^	1.112	<0.001
ADF (%)	31.69 ^a^	28.92 ^a^	22.34 ^b^	1.198	0.002
Ca (%)	40.01 ^a^	38.31 ^a^	34.17 ^b^	0.988	0.024
P (%)	48.80 ^a^	46.26 ^a^	39.64 ^b^	0.820	<0.001

HP, high crude protein; MP, medium crude protein; LP, low crude protein; SEM = standard error of least square means; ^a, b^ means within the same row not followed by the same letters are significantly different at *p* < 0.05.

**Table 4 animals-12-02066-t004:** Effects of dietary crude protein levels on serum metabolites of lactating donkeys.

Items	HF	WF	LP	SEM	*p*-Value
TP, g/L	66.70 ^a^	62.38 ^a^	57.08 ^b^	1.505	0.010
ALB, g/L	35.21 ^a^	34.41 ^a^	29.49 ^b^	1.067	0.011
BUN, mmol/L	8.89 ^a^	8.31 ^a^	7.10 ^b^	0.251	0.002
ALP, U/L	176.30	174.08	188.42	4.542	0.112
GLU, mmol/L	5.18 ^a^	5.02 ^ab^	4.71 ^b^	0.107	0.018
NEFA, umol/L	0.17 ^b^	0.19 ^b^	0.28 ^a^	0.017	0.002
BHBA, mmol/L	0.36 ^b^	0.39 ^b^	0.43 ^a^	0.011	0.013
TC, mmol/L	1.85	1.88	2.05	0.090	0.264
Ca, mg/L	2.73	2.78	2.78	0.059	0.848
P, mg/L	1.29	1.33	1.30	0.045	0.778

HP, high crude protein; MP, medium crude protein; LP, low crude protein; SEM = standard error of least square means; ^a, b^ means within the same row not followed by the same letters are significantly different at *p* < 0.05.

## Data Availability

The data presented in this study are available upon request from the corresponding author.

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
