# Peer review of "Effect of Varying Dietary Crude Protein Level on Milk Production, Nutrient Digestibility, and Serum Metabolites by Lactating Donkeys"

_animals, 2022, doi:10.3390/ani12162066_

Round 1

Reviewer 1 Report

Overall, this is a well-developed study examining the effects of different protein levels on milk production in donkeys. The authors provide a clear rationale for conducting the study, concisely explain their methodology, and discuss their results in relation to studies of other livestock/animal taxa. Most of my comments are minor. Please see below.

Line 16: change “researches” to “research”

Line 22: The abstract could be improved by providing a sentence about why the study was conducted. For example, what gap in the literature is this study addressing? Some of this information is provided in the Simple Summary, but it would also be nice to have some of this information in the abstract.

Line 33: Change “In a word” to “In short”

Line 43: remove “in the present”

Line 43-44: This sentence is not clear to me. Please consider re-wording.

Line 46: Consider changing “maternal” to “human”

Line 51-53: Change wording to “Compare with ruminant milk, donkey milk generally contains higher levels of whey protein and lysozyme which are known for their antioxidative and antibacterial properties.”

Line 54-55: Do the authors have values to compare milk yield of ruminants vs. donkeys?

Line 58: Are there any citations to support this “intense interest”?

Line 70: remove “by nutritional strategy”

Line 102: Consider changing “eye contact” with “visibility”

Methods section: In general, this section is well-written and clearly defined.

Lines 149-157: Was individual donkey controlled for in anyway (e.g., as a random/repeated variable)?

Lines 157: Did you use adjusted p-values given the multiple pairwise comparisons?

Table 2: It would be helpful to have more information in the title. Are these values estimated marginal means or means? It would be helpful to have standard deviations or confidence intervals to assess the range of variation in the values.

Line 201: If the authors mention “few” studies on this topic, why do they not cite them?

Lines 200-228: Nice discussion of how the results of this study generally corroborate results of other studies on other taxa. This section is lacking information/explanation on why milk yield increases with CP increase but only up to a certain point. Do the authors have an explanation for why that is?

Lines 305-307: Considering changing to “Future research is required on the effects of dietary CP level on mammary gland development in donkeys.

Reviewer 2 Report

Review of Yue et al.

Overall, a well constructed, analyzed, and written experiment. I found it easy to follow. I have a few comments.

1)      There are some slight English language issues that should be caught by the copy editor

2)      In Lines 104-111 you use equations from dairy cows to calculate these parameters. Could you comment on any limitations of apply equations across species? Could add a statement to the methods here or to the discussion.

3)      Line 223-4 This sentence is not clear, please rephrase.

Reviewer 3 Report

Please find enclosed a file containing the comments and suggestions for Authors.
